# Predominance of the SARS-CoV-2 Lineage P.1 and Its Sublineage P.1.2 in Patients from the Metropolitan Region of Porto Alegre, Southern Brazil in March 2021

**DOI:** 10.3390/pathogens10080988

**Published:** 2021-08-05

**Authors:** Vinícius Bonetti Franceschi, Gabriel Dickin Caldana, Christiano Perin, Alexandre Horn, Camila Peter, Gabriela Bettella Cybis, Patrícia Aline Gröhs Ferrareze, Liane Nanci Rotta, Flávio Adsuara Cadegiani, Ricardo Ariel Zimerman, Claudia Elizabeth Thompson

**Affiliations:** 1Graduate Program in Cell and Molecular Biology (PPGBCM), Center of Biotechnology, Universidade Federal do Rio Grande do Sul (UFRGS), Porto Alegre 91501-970, RS, Brazil; vinicius.franceschi@ufrgs.br; 2Graduate Program in Health Sciences, Universidade Federal de Ciências da Saúde de Porto Alegre (UFCSPA), Porto Alegre 90050-170, RS, Brazil; gabrield@ufcspa.edu.br (G.D.C.); p.ferrareze@gmail.com (P.A.G.F.); lnrotta@gmail.com (L.N.R.); 3Department of Infection Control and Prevention, Hospital da Brigada Militar, Porto Alegre 91900-590, RS, Brazil; drchristianoperin@gmail.com (C.P.); amariantehorn@gmail.com (A.H.); 4Laboratório Exame, Novo Hamburgo 93510-250, RS, Brazil; camilapptr@gmail.com; 5Department of Statistics, Universidade Federal do Rio Grande do Sul, Porto Alegre 91501-970, RS, Brazil; gcybis@gmail.com; 6Corpometria Institute, Brasília 70390-150, DF, Brazil; flavio.cadegiani@unifesp.br; 7Department of Pharmacosciences, Universidade Federal de Ciências da Saúde de Porto Alegre, Porto Alegre 90050-170, RS, Brazil

**Keywords:** COVID-19, severe acute respiratory syndrome coronavirus 2, infectious diseases, high-throughput nucleotide sequencing, molecular evolution, molecular epidemiology, phylogeny

## Abstract

Almost a year after the COVID-19 pandemic had begun, new lineages (B.1.1.7, B.1.351, P.1, and B.1.617.2) associated with enhanced transmissibility, immunity evasion, and mortality were identified in the United Kingdom, South Africa, and Brazil. The previous most prevalent lineages in the state of Rio Grande do Sul (RS, Southern Brazil), B.1.1.28 and B.1.1.33, were rapidly replaced by P.1 and P.2, two B.1.1.28-derived lineages harboring the E484K mutation. To perform a genomic characterization from the metropolitan region of Porto Alegre, we sequenced viral samples to: (i) identify the prevalence of SARS-CoV-2 lineages in the region, the state, and bordering countries/regions; (ii) characterize the mutation spectra; (iii) hypothesize viral dispersal routes by using phylogenetic and phylogeographic approaches. We found that 96.4% of the samples belonged to the P.1 lineage and approximately 20% of them were assigned as the novel P.1.2, a P.1-derived sublineage harboring signature substitutions recently described in other Brazilian states and foreign countries. Moreover, sequences from this study were allocated in distinct branches of the P.1 phylogeny, suggesting multiple introductions in RS and placing this state as a potential diffusion core of P.1-derived clades and the emergence of P.1.2. It is uncertain whether the emergence of P.1.2 and other P.1 clades is related to clinical or epidemiological consequences. However, the clear signs of molecular diversity from the recently introduced P.1 warrant further genomic surveillance.

## 1. Introduction

After its initial emergence in Wuhan (China) in late 2019, severe acute respiratory syndrome coronavirus 2 (SARS-CoV-2) spread rapidly around the world leading to the COVID-19 pandemic officially recognized in March 2020 [1]. As of May 17, 2021, >163 million cases and >3.3 million deaths have been confirmed. In Brazil, the third most affected country by COVID-19, >15.6 million cases and >435,000 deaths have been reported. From a virological standpoint, this could be related to the continental magnitude of Brazil, leading to multiple viral introductions [2] and the recent emergence of a novel variant of concern (VOC) presenting enhanced infectiousness.

Rio Grande do Sul (RS) is the southernmost state in Brazil. It is bordered southerly by Uruguay, westerly by Argentina, and northerly by the state of Santa Catarina, Brazil. With an estimated population of 11.5 million inhabitants and 39.79 inhabitants per square kilometer, RS is the 6th most populous and the 13th most densely populated state in the country [3]. Since 2017, the Brazilian Institute of Geography and Statistics (IBGE) has divided RS into eight intermediate geographic regions: Porto Alegre, Pelotas, Uruguaiana, Santa Maria, Santa Cruz do Sul/Lajeado, Ijuí, Passo Fundo, and Caxias do Sul [4]. The municipality of Porto Alegre is the state capital, and its metropolitan region comprises 34 municipalities aggregating >4 million inhabitants (~2% of the country’s population) and is characterized by intense transit of people. COVID-19 was firstly confirmed in RS on 10 March 2020, in a returning traveler from Italy [5]. The state implemented in May 2020 the “controlled distancing system”, which divided the state into 21 regions and 26 areas (R01–R26) and consisted of a flag system establishing restrictions and flexibilizations of non-essential activities based on the weekly occupation of intensive care unit beds and expected deaths. However, due to economic losses, the more amenable “shared management system” allowed mayors to appeal in court and adopt less restrictive flag protocols [6].

Important shifts of COVID-19 epicenters have occurred during 2020, starting with Asia and followed by Europe, North America, and South America. After months of relatively slow evolution, novel VOCs (e.g., B.1.1.7, B.1.351, P.1, and B.1.617.2) harboring a constellation of signature mutations in the spike protein have emerged [7]. More recently, the World Health Organization (WHO) assigned labels for the VOCs based on Greek letters. Therefore, the four VOCs are named, respectively, Alpha, Beta, Gamma, and Delta. These lineages independently arose in the United Kingdom [8], South Africa [9], Brazil [10,11], and India [12,13] and have fueled secondary outbreaks in places where they have emerged, despite previous rates of seroprevalence of up to 75% [14]. The city of Manaus (Amazonas, Brazil), the probable place of origin of the P.1 lineage, faced a major second wave of COVID-19. An explosive resurgence of cases and deaths became evident in mid-December 2020. Since the P.1 variant carries multiple mutations of potential biological significance (especially E484K, K417T, and N501Y in the receptor-binding domain (RBD) of the spike protein): (i) some key substitutions may lead to the immunity evasion; (ii) higher transmissibility when compared with pre-existing lineages has been characterized; (iii) this VOC has been the focus of increased surveillance and deserves being studied in greater detail [15]. After this outbreak, almost all Brazilian states experienced increases in the number of cases, hospitalizations, intensive care unit (ICU) admissions, and deaths, resulting in a reemergence of the public health crisis previously experienced in the first wave of COVID-19 [16].

The diversity of SARS-CoV-2 during the first epidemic wave in Brazil was mainly composed of B.1.1.28 and B.1.1.33 lineages [2,17], although the very low sequencing rate across the country has limited these estimates [17]. However, these previous lineages were rapidly replaced by P.1 and P.2 in late 2020 and early 2021, which are both derived from the common ancestor B.1.1.28 and harbor concerning mutations in the spike protein (e.g., E484K and N501Y) [17,18]. In RS state, the most common lineages identified up to May 2021 are still B.1.1.33 (n = 290) and B.1.1.28 (n = 238) [19,20]. Nevertheless, P.1 has emerged as the most prevalent lineage sequenced in more recent samples [19]. Recently, newer mutations were detected in addition to the original set presented in P.1, giving rise to the sublineage P.1.2 [21]. P.2 probably emerged in Rio de Janeiro state (Southeast) [22], but it was also found in several municipalities of RS state as of October 2020 [23,24]. The first P.1 infection in the state was once thought to be in a patient from Gramado city in February 2021 [25]. However, in a more recent study, the actual first P.1 was detected on 30 November 2020. This happened in a patient with comorbidities from Campo Bom city, who was reinfected by the P.2 lineage on 11 March 2021 [26].

Even though RS was one of the least affected Brazilian states in the first epidemic wave, it suffered a pronounced increase in cases in late 2020 [16]. In February 2021, the progressive increases in cases and hospitalizations (3.8-fold) led to the collapse of the local state healthcare system. Since recent findings of the widespread dissemination of the SARS-CoV-2 lineage P.1 in Brazil have been confirmed, we sequenced samples from patients from the metropolitan region of Porto Alegre to: (i) identify the prevalence of SARS-CoV-2 lineages in the region, the state, and bordering countries/regions; (ii) characterize the mutation spectra; (iii) hypothesize possible viral dispersal routes by using phylogenetic and phylogeographic approaches.

## 2. Results

### 2.1. Epidemiological Information

Of the 56 samples of hospitalized patients between March 9 and 17 2021, 75.0% (n = 42) of them were male, and the mean age was 37.2 years (interquartile range (IQR): 13.5 years). The mean cycle threshold (Ct) value for the first RT-qPCR conducted at Laboratório Exame was 19.12 cycles (median: 18.00; IQR: 6.00 cycles). Forty-seven (83.9%) had contact with a confirmed or suspected case. The majority of them were from the RS state capital, Porto Alegre (n = 32; 57.1%). In total, 51 (91.1%) were from the intermediate geographic region of Porto Alegre and 5 (8.9%) from the intermediate region of Santa Maria (Table 1 and Appendix A).

### 2.2. SARS-CoV-2 Mutations and Lineages

Consensus SARS-CoV-2 genomes were obtained with an average coverage depth of 813.2× (median: 820.6×; IQR: 184.7×) (Appendix A). We detected 175 different mutations comprising all samples (Figure 1A). The ORF1ab carried 102 (58.3%) replacements followed by spike (n = 24; 13.7%), nucleocapsid (n = 18; 10.3%), ORF3a (n = 14, 8.0%), ORF7a (n = 6; 3.43%), ORF8 (n = 5; 2.86%), and membrane (n = 3; 1.7%) genes. Remarkably, 50% of the spike substitutions occurred in only one genome, and, of these, nine (75.0%) were missense (Appendix A). Fifty-nine (33.7%) mutations were identified in two or more sequences. From these, 36 (61.0%) are non-synonymous (missense), 21 (35.6%) are synonymous, 1 (1.7%) is intergenic at 5′ Untranslated Region (UTR), and 1 (1.7%) is an inframe deletion. Highly frequent (≥10 genomes) mutations were found in 34 genomic positions, 24 (70.5%) being missense and 9 (26.5%) synonymous. Fifteen substitutions (10 in the spike protein: L18F, T20N, P26S, D138Y, R190S, K417T, E484K, N501Y, H655Y, and T1027I) are P.1 lineage-defining mutations (Figure 1B and Table 2). The only P.1 defining replacement not found at high frequency in our study was the deletion in ORF1ab (del 11288:11296), called in only four genomes. Deletions overlapping annealing sites of amplicon primers are associated with a strong decrease in the PCR efficiency prior to sequencing, leading to low genomic coverage [27]. Then, after applying a stringent coverage depth filter (DP > 50) for calling the genomic positions in the consensus sequences, this deleted region was flagged as low coverage. 

Most importantly, other positions presenting single nucleotide polymorphisms (SNPs) reached the appropriate threshold, since a point mutation is generally unable to cause dropouts.

After comparing the frequency of mutations from the recently sequenced samples and the Brazilian P.1 genomes, we observed a combination of mutations that stood out in a significant proportion (n = 11; 19.6%) compared with previous P.1 sequences from Brazil. This combination was previously described [21] and gave rise to the P.1.2 lineage, which harbors three ORF1ab replacements (synC1912T, D762G, and T1820I), one in ORF3a (D155Y), and one in N protein (synC28789T) (Table 2). Additionally, two of these genomes (18.2%) carry T11296G (ORF1ab nsp6: F3677L) and eight (72.7%) harbor G25641T (ORF3a: L83F) substitutions. Another cluster, made of four local genomes and subsequently named Clade 2, was also detected. This clade possesses three defining mutations (ORF1ab nsp4: V2862L, synC10507T, and ORF3a: M260K), but it does not fall into a lineage designation at this moment but deserves further monitoring (Appendix A).

Considering PANGO lineages, 54 genomes (96.4%) were designated as P.1, one (1.8%) as P.2, and one (1.8%) as B.1.1.28. Even without being classified according to the Pango-designation’s most updated version, the P.1.2 lineage was present in 11/54 (20.4%) of the P.1 sequences (https://github.com/cov-lineages/pango-designation/issues/56; accessed on 6 May 2021) (Appendix A).

### 2.3. Lineage Distribution in Neighboring Countries and Brazilian Regions

RS state shares borders with Argentina to its west (Appendix A), leading to the transit of people at the frontiers. From March to April 2020, B.1 was the most prevalent lineage in this bordering country. B.1.499 and N.3 were abundant from May to July when the N.5 started to rise and surpassed B.1.499 in November 2020 (Figure 2A). Importantly, N.3 and N.5 are derived from the B.1.1.33 lineage widespread in Brazil (https://cov-lineages.org/lineages.html; accessed on 4 May 2021). The P.2 lineage, which initially emerged in Brazil [22] and derived from another Brazilian disseminated lineage (B.1.1.28), was firstly found in November 2020, and, by January and February 2021, it had already outnumbered the other lineages in Argentina (Figure 2A).

In the entire Brazil, despite early introductions of B.1 and B.1.1, lineages B.1.1.28 and B.1.1.33 were most abundant from March to October 2020. In October, P.2 already represented an important portion of the sequences, and, by November, it had already surpassed B.1.1.33. In December 2020 and January 2021, with the emergence of P.1, this lineage and P.2 already became the most prevalent, while, between February and April, P.1 replaced all other lineages (Figure 2A,B). Some fluctuations evidently occurred in different Brazilian regions, such as a prevalence of more local lineages (e.g., B.1.195 and B.1.1.378 in the Northern region, B.1.1 and N.9 in the Northeast, and B.1.1.7 in the Southeast and Centre-West regions) (Appendix A). In RS, a similar landscape was observed compared to the Brazilian scenario. B.1.1.28 and B.1.1.33 were most prevalent until October 2020, when P.2 emerged and remained until January 2021 along with B.1.1.28 as the most prevalent. After the introduction of P.1 (January 2021), this lineage practically supplanted the others in February and March 2021 (Figure 2A,B).

After dividing RS into the intermediate regions proposed by IBGE (Appendix A), it was possible to gain insights into the dynamics of the lineages in the state, despite the low sample size of some regions (Figure 2C). In most regions, the lineages B.1.1.28 and B.1.1.33 were more prevalent, but P.2 was also detected. In fact, in the Caxias do Sul region, more P.2 (n = 31; 49.2%) were sequenced in relation to other lineages. Since the Porto Alegre region has a larger sample size, we divided its results by year to check the most recent (2021) evolutionary abundance. In 2020, B.1.1.33 (n = 229; 49.3%) and B.1.1.28 (n = 137, 29.5%) were the most abundant, followed by P.2 (n = 37; 8.0%). In 2021, P.1 (n = 26, 68.4%) and P.2 (n = 6, 15.8%) have already outperformed the other lineages (Figure 2C). In our study from March 2021, 96.4% of the samples were classified as P.1. We were able to identify a new P.1 sublineage (P.1.2) in 11 (20.4%) genomes from four different municipalities (Porto Alegre, Canoas, São Sebastião do Caí, and Santa Maria), demonstrating the possible diversification of P.1 and its spread within RS (Figure 2C and Appendix A).

### 2.4. Maximum Likelihood Phylogenomic Analysis

After running the Nextstrain workflow using quality control and subsampling approaches, we obtained a dataset of 8635 time- and geographical-representative genomes. From these, 861 were from Africa, 1370 from Asia, 2219 from Europe, 481 from North America, 218 from Oceania, and 3486 from South America. Brazil was represented by 2608 sequences and RS state by 730 sequences (56 from this study and 674 available in GISAID) (Appendix A).

The time-resolved ML phylogenetic tree confirmed the PANGO lineages assigned, since 54 genomes (96.4%) grouped with P.1 representatives, 1 (1.8%) with B.1.1.28, and 1 (1.8%) with P.2 sequences. We also observed a strong correlation between genetic distances and sampling dates (R^2^ = 0.71). The P.1 sequences were grouped above the regression line, showing higher evolutionary rates than the other lineages in the SARS-CoV-2 phylogeny, as observed in other studies [11,14]. We highlighted the most abundant global lineages present in RS state that passed the quality control criteria (B.1.1 (n = 32), P.1 (n = 83), P.2 (n = 83), B.1.1.28 (n = 203), and B.1.1.33 (n = 286)). We also noticed the high abundance of B.1.1.28 and B.1.1.33 lineages until October and November 2020, followed by the rise and establishment of P.2 and P.1, respectively (Figure 3A). 

The only B.1.1.28 sequence identified in this study (RS-HBM-39491) branched in a clade represented by 30 sequences, mostly represented by Southeastern Brazilian (n = 17; 56.6%) genomes. This clade is supported by the ORF1ab:synC15810T mutation and includes a subclade characterized by the ORF1ab:L4182F mutation, where the local sequence is placed together with four samples from São Paulo (SP), one from Portugal, and one from Chile (Appendix A). Most importantly, this local genome harbors seven other mutations: ORF1ab:T2087I (nsp3), D3022N (nsp4), N3970S (nsp8), V4436A (RNA-dependent RNA polymerase), synC13724T, synG18973A, and intergenic:G29736T. Additionally, the only P.2 sequence from this study (RS-HBM-39486) formed a separate clade composed of 20 sequences from several Brazilian states (10 from RS, 1 from Paraná, 1 from SP, and 1 from Maranhão), 5 from Argentina, 1 from USA, and 1 from Norway (Appendix A). This clade is characterized by the ORF1ab:synT6218C (nsp3) mutation. Moreover, this local sequence accrued seven specific mutations: ORF1ab:synA7201G, S2926F (nsp4), V6871A (2′-O-ribose methyltransferase), S:G1251V, ORF7a:G38V, N:synC28333T, and intergenic:G29688T.

To get a more detailed understanding of the P.1 diffusion throughout Rio Grande do Sul, other Brazilian regions, and worldwide, we built an ML tree of 4499 genomes belonging to this lineage (Appendix A). P.1 sequences from this study were allocated into several distinct branches, suggesting multiple introductions and the formation of different P.1-derived clades and clusters. 

We identified 4 clades, 5 clusters, and 13 isolated sequences (Figure 3B and Appendix A). Most importantly, Clade 1 had high branch-support (SH-aLRT = 76.3) and was composed of 11 sequences originated in this study that shared five lineage-defining mutations as previously described (Table 2) and were recently attributed to the P.1.2 sublineage (https://github.com/cov-lineages/pango-designation/issues/56; accessed on 6 May 2021). As of April 26, 2021, this sublineage is already distributed worldwide in 93 sequences (the Netherlands, Spain, England, and USA) and in other Brazilian states (Rio de Janeiro (RJ) and SP) [21]. Clade 2 sequences harbored two mutations in ORF1ab:V2862L (nsp4) and synC10507T and one in ORF3a:M260K, and it comprised 81 genomes. Four samples are from this study. The majority are from Amazonas (n = 15), São Paulo (n = 11), RS (n = 8), and Bahia (BA) (n = 4), and worldwide sequences are mainly represented by French Guiana, USA, Spain, Japan, and Jordan. Clade 3 is represented by three ORF1ab mutations (synC1420T, D1600N [nsp3], and synT8392A) in three of the seven local genomes. It is composed of sequences from RS (n = 25), SP (n = 15), Maranhão (n = 10), and RJ (n = 8), as well as other countries (mainly Spain, French Guiana, and USA). Clade 4 is characterized by two ORF1ab substitutions (G400S (nsp2) and S6822I (2′-O-ribose methyltransferase)), one N:synT26861C in three genomes, and carries other additional mutations (ORF1ab: synG10096A, G3676S (nsp6), F3677L (nsp6)) and M:synT26861C. This clade is mainly found in SP (n = 11), RS (n = 7), Santa Catarina (n = 5), BA (n = 4), and Goiás (n = 4), as well as other countries (mainly USA, Chile, and England).

Clusters 1 and 3 have, respectively, one (ORF1ab: G3676S (nsp6)) and two (ORF1ab: synC1471T and A1049V (nsp3)) shared mutations. Among all identified clusters, the most diverse was Cluster 5, which contains three samples from this study and has five defining mutations: four in ORF1ab (synT4705C, synC11095T, syn11518, and T5541I (helicase)) and one in ORF7a: E16D. Moreover, two sequences share one distinct mutation (ORF1ab: F3677L (nsp6)).

### 2.5. Bayesian Molecular Clock and Phylogeographic Analysis

To date the time of the most recent common ancestor (TMRCA) and the diffusion of the four P.1 clades identified in our ML analysis, we used coalescent and phylogeographic methods. For Clade 1, which is correspondent to the recently labeled P.1.2 lineage, sequences showed a moderate correlation of genetic distances and sampling dates (correlation coefficient: 0.59, R^2^ = 0.34) (Figure 4A). We estimated a median evolutionary rate of 7.68 × 10^−4^ (95% highest posterior density interval [HPD]: 4.18 × 10^−4^ to 1.14 × 10^−3^ subst/site/year) and the TMRCA on 18 December 2020 (95% HPD: 29 October 2020 to 31 January 2021). Interestingly, the tree’s root was placed in RS, between a sequence from RS (the oldest sequence from this clade: EPI_ISL_983865) and a subclade from USA. The divergence of these subclades was dated on 15 January 2021 (95% HPD: 15 January to 26 March 2021). The subclade composed of the RS sequences formed two separate clusters, one with three sequences from this study and one Australian genome and another composed of sequences from RS, SP, UK, Portugal, USA, and transmission clusters from RJ and Netherlands (Figure 4B). The emergence of an important cluster in RJ carrying additional mutations [21] was dated here on 11 March 2021 (95% HPD: 11 March to 6 April 2021). As American sequences formed a separate subclade, local transmission is probably occurring in the country. The divergence of the American subclade was dated 7 February 2021 (95% HPD: 1 February to 11 May 2021). In accordance with the root being placed in RS, the BSSVS procedure identified well-supported rates of diffusion from RS to other Brazilian states such as São Paulo (Bayes Factor (BF): 6.82; posterior probability (PP): 0.52), Rio de Janeiro (BF: 39.18; PP: 0.86), and other countries such as USA (BF: 31.94; PP: 0.84) and Netherlands (BF: 80.16; PP: 0.93). 

However, it is possible that this lineage emerged in another Brazilian state, but its earlier representatives were not sampled. This is a strong hypothesis since this sequence is associated with community transmission after contact with tourists in a city of RS (Gramado) that receives numerous visitors annually [25].

For Clade 2, we estimated a median evolutionary rate of 5.85 × 10^−4^ (95% HPD: 4.18 × 10^−4^ to 7.71 × 10^−4^ subst/site/year), and the TMRCA was dated 30 November 2020 (95% HPD: 2 November to 21 December 2020). This clade includes sequences from 11 Brazilian states from all 5 regions and 9 other countries. We were able to detect at least five introductions from Amazonas, where this clade probably emerged. These introductions ranged from 28 December 2020 (95% HPD: 28 December 2020 to 5 January 2021) to January 28, 2021 (95% HPD: 28 January to 7 March 2021). Importantly, we identified a well-supported subclade (PP = 1) of four genomes from this study (Figure 5A).

For Clade 3, the TMRCA was estimated on 20 December 2020 (95% HPD: 25 November to 29 December 2020) and the median evolutionary rate was 7.85 × 10^−4^ (95% HPD: 6.06 × 10^−4^ to 1.02 × 10^−3^ subst/site/year). This clade harbors sequences from 9 Brazilian states and 10 other countries. Amazonas is the most probable source of its emergence. From then onwards, multiple transmission clusters were established in foreign countries (e.g., Spain, Portugal, and USA) and Brazilian states (especially Maranhão, SP, and RS). This clade was introduced at least 5 times in RS, leading to 2 major subclades represented by 18 and 4 sequences, respectively. The major subclade (n = 18, PP = 0.98) was dated 11 January 2021 (95% HPD: 11 January to 1 February 2021) (Figure 5B).

For Clade 4, the TMRCA was dated on 2 December 2020 (95% HPD: 7 October 2020 to 3 January 2021), and the median evolutionary rate was 6.26 × 10^−4^ (95% HPD: 3.51 × 10^−4^ to 1.01 × 10^−3^). This clade comprises nine Brazilian states and five foreign countries. After its initial emergence and spread in Amazonas, it had already formed transmission clusters in SP, BA, United Kingdom, and USA. Most importantly, a subclade containing sequences from two neighboring states from Southern Brazil (seven from RS and five from Santa Catarina (SC)) indicates its diffusion from RS to SC, probably leading to two separate introductions. The divergence of this subclade was estimated on 16 December 2020 (95% HPD: 16 December 2020 to 19 January 2021) (Figure 5C).

Phylogenetic and molecular clock approaches suggest the wide circulation of the VOC P.1 both nationally and internationally between late 2020 and early 2021. This lineage has already diversified into some clades that bear characteristic mutations, although they exhibit similar evolutionary rates. We inferred that P.1 (and its derived clades) was introduced multiple times in the southernmost Brazilian state (RS) between mid-December 2020 and January 2021. Remarkably, this date is close to the first P.1 detection in Manaus, which is located ~4000 km away. These early introductions led to the formation of local subclades that could be identified even using a reduced set of sequenced samples.

## 3. Discussion

In this study, the analysis of 56 samples from the state of Rio Grande do Sul (RS), Southern Brazil, confirmed that the P.1 lineage was already highly prevalent. Interestingly, we demonstrated that P.1 is already showing signs of diversification and has originated a new sublineage (P.1.2). Herein, we indicate the likely origin and the first clusters of this novel lineage. This sublineage was detected in three Brazilian states, and other countries, and its most recent common ancestor was dated on mid-December, 2020 (95% HPD: 29 October 2020 to 31 January 2021). In accordance with the majority of the states from Brazil, this state experienced significant increases in hospitalizations in early 2021. This scenario was related to the emergence and rapid spread of the P.1 variant across the country.

After almost one year of relatively slow SARS-CoV-2 evolution, the emergence of multiple and convergent lineages harboring a constellation of mutations in the spike protein raised concern in the scientific community. This protein is responsible for mediating interaction with the human Angiotensin-Converting Enzyme 2 receptor (hACE2) and is a primary target of neutralizing antibodies and vaccines [28]. The variants harboring different mutational signatures, including spike protein substitutions, were classified as VOCs and “variants of interest” (VOIs), depending on their growing relevance in the current pandemic. The first three VOCs emerged in England (B.1.1.7) [8], South Africa (B.1.351) [9], and Brazil (P.1) [10]. More recently, B.1.617.2 (India) [12,13] also was characterized as a VOC. By May 2020, B.1.427/429 [29], B.1.526 (New York, USA), B.1.617 (India), and P.2 (Brazil) [22] were categorized as VOIs. B.1.1.7, B.1.351, and P.1, the most studied VOCs, have the D614G and N501Y mutations in common. B.1.351 and P.1 share a mutation in the K417 site (K417N and K417T, respectively) and the E484K replacement, which is also observed in the P.2 lineage. Additionally, B.1.1.7 carries the P681H substitution in the furin-cleavage site and multiple VOIs bear the L452R substitution [7]. The presence of common substitutions in different SARS-CoV-2 lineages suggests co-evolutionary and convergent mutational processes [8,9,10,30].

In the present study, we noticed that B.1.1.33 and B.1.1.28 lineages, detected at the beginning of the pandemic in Brazil [2], had been similarly prevalent in different regions until September 2020, before the appearance of P.2 (in October) and P.1 (in December 2020). The B.1.1.33 lineage shows variable abundance in different Brazilian states (ranging from 2% in Pernambuco to 80% in Rio de Janeiro), with moderate prevalence in South American countries (5–18%). Surprisingly, this lineage was firstly detected in early March 2020 in other American countries (e.g., Argentina and USA). Apparently, an intermediate strain probably emerged in Europe and subsequently spread to Brazil, where its spread gave rise to B.1.1.33 [31] and possibly triggered secondary outbreaks in Argentina and Uruguay [31,32]. We found that N.3 and N.5, both derived from B.1.1.33, represented an important proportion of the sequences from Argentina from May to December 2020, when it was replaced by the P.2 lineage, which probably emerged in Rio de Janeiro (Southeastern Brazil). The B.1.1.28 lineage, despite apparently being less abundant than B.1.1.33 in several Brazilian regions, quickly diversified into two variants: VOC P.1 and VOI P.2 [33]. Since the end of 2020, these two lineages have lead the diversity of SARS-CoV-2 in Brazil [17] and have caused concern in other countries after several introductions. Regarding the distribution of sequenced samples across RS state, the cumulative frequency of B.1.1.28 and B.1.1.33 was higher until mid-April 2021 [16]. However, since the end of 2020 and beginning of 2021, a rise in P.1 and P.2 sequences was observed. Our study supported that P.1 outperformed other lineages in RS as of March 2021, although the collection of samples in hospitalized patients and low geographic representativeness does not allow the extrapolation of these findings.

The emergence of a B.1.1.28 derived lineage carrying the S:E484K mutation (P.2) was dated, in a retrospective study, late February 2020 in the Southeast (São Paulo and Rio de Janeiro), followed by transmission to the South (especially RS). Since then, multiple dispersion routes were observed between Brazilian states, especially in late 2020 and early 2021 [18]. However, this lineage went unreported until October 2020, when it was first detected in the state of Rio de Janeiro [22] and in the small municipality of Esteio in RS [23]. The increased frequency of B.1.1.28 and derived lineages was corroborated by another study that included samples from several municipalities of RS in November 2020. This study found that 86% of the genomes could be classified as B.1.1.28 and ~50% of these, in fact, belong to the new lineage P.2 [24]. Nonetheless, our current study suggests that P.2 has already been nearly entirely replaced by the P.1 lineage or is not particularly well represented among the analyzed patients seeking emergency consultation or requiring hospitalization.

Between June and October 2020, an extremely high seroprevalence (44–76%) was observed in Manaus (Amazonas, Brazil) in a study from blood donors [14]. However, despite these numbers, Manaus faced a resurgence of cases and a six-fold increase in hospitalizations between December 2020 and January 2021. The most plausible hypotheses that would justify this condition are: (i) the previous overestimation of seroprevalence in Manaus; (ii) the immune evasion property of some SARS-CoV-2 mutations found in VOCs; (iii) higher transmissibility and pathogenicity of SARS-CoV-2 lineages circulating in the second wave compared with pre-existing lineages [15].

A genomic epidemiology study that used 250 SARS-CoV-2 genomes from 25 different municipalities from Amazonas sampled between March 2020 and January 2021 shows that the first exponential phase in the state was driven mainly by the spread of lineage B.1.195, which was gradually replaced by B.1.1.28. The second wave coincided with the emergence of P.1 in November, which rapidly replaced the parental lineage (<2 months) [11] and whose emergence was preceded by a period of rapid molecular evolution [10]. Importantly, rapid accumulation of mutations over short timeframes have been reported in chronically infected or immunocompromised hosts [34,35]. However, preliminary findings pointed to the existence of P.1 intermediate lineages, suggesting that the constellation of mutations defining P.1 were acquired at sequential steps during multiple rounds of infections instead of within a single long-term infected individual [36]. The VOC P.1 carries three deletions, four synonymous substitutions, a four base-pair nucleotide insertion, and at least 17 other lineage-defining replacements, including 10 missense mutations in the spike protein (L18F, T20N, P26S, D138Y, R190S, K417T, E484K, N501Y, H655Y, and T1027I), 8 of which are subjected to positive selection [10]. 

Regarding infectiousness, transmissibility, and case fatality, the viral load was ~10-fold higher in P.1 infections than in non-P.1 infections [11]. Although another study points to uncertainties regarding viral load and duration of infection after accounting for confounding effects [10]. Moreover, it was estimated to be 1.7–2.4-fold more transmissible, raising the probability that reinfections would be caused more frequently in hosts infected by P.1 rather than by older lineages. Remarkably, infections were 1.2–1.9 times more likely to result in death in the period following the emergence of P.1 compared to previous time frames [10]. These findings support that successive lineage replacements in Amazonas were driven by a complex combination of factors, including the emergence of the more transmissible VOC P.1 virus [11]. 

A study conducted in RS described a P.1 lineage infection on 30 November 2020 followed by a P.2 lineage reinfection on 11 March 2021 in a patient with comorbidities. This report was the first detected P.1 in the state [26]. Other analyses suggest that the P.1 lineage presumably emerged in Manaus, Brazil, in mid-November 2020 [10,11]. Therefore, the P.1 lineage was present in Southern Brazil about a few days after its emergence in Manaus, Northern Brazil. Our molecular clock analysis supported this scenario. Another study, once thought to be the first P.1 report in RS, documented local transmission of P.1 from a person who had close contact with tourists and was positive for COVID-19 in early February 2021 [25]. This happened in the city of Gramado, a town in the mountains that receives around 6.5 million tourists every year and belongs to the Caxias do Sul intermediate region. Interestingly, this sample from Gramado was the earliest representative of a new P.1-derived lineage (P.1.2), described in 11 patients from our study and found in transmission clusters from the RJ state in Southeastern Brazil, USA, and the Netherlands. Remarkably, our local sequences are more similar to genomes from other countries compared to the RJ cluster, which acquired at least four additional mutations (including S:A262S) [21].

Whether P.1.2 has worse clinical outcomes than its prior variant (P.1) is unknown. However, as described above, the missense mutations characteristic of the new sublineage are located at nsp2 and nsp3 (ORF1ab), ORF3a, and nucleocapsid. These sites are known for their interaction with human proteome, potentially influencing the immunological and inflammatory response against SARS-CoV-2 infection [37]. The ORF3a:D155Y substitution is located near SARS-CoV caveolin-binding Domain IV. The binding interaction of viral ORF3a protein to host caveolin-1 is essential for entry and endomembrane trafficking of SARS-CoV-2. Since this mutation breaks the salt bridge formation between Asp155 and Arg134, it can interfere with the binding affinity of ORF3a to host caveolin-1 and change virulence properties. Most importantly, this disrupted interaction may be associated with improved viral fitness, since it can avoid the induction of host cell apoptosis or extend the asymptomatic phase of infection [38]. We hypothesize that these new substitutions could, therefore, influence epidemiological and clinical outcomes favoring P.1.2 evolution. This is elusive at best at this time, however, and further sublineage characterization is needed to further explore its real relevance.

Some limitations should be considered. Firstly, the sample size is low and not necessarily representative of RS state. Considering the number of sequences from each intermediate region in RS available in GISAID, it is very likely that the distribution seen on the map (Figure 2C) is a consequence of sampling at different times in these localities or simple randomness. Thus, it should not be assumed as a true representation of the spatial diversity in the state. Since publicly available genomes are a result of episodic sequencing efforts, especially in Brazil, more precise inferences about introductions and diffusion processes in regional and worldwide contexts are restricted due to the lack of proper geographical and temporal distribution of the samples. Therefore, more research and surveillance are essential to unravel a more precise genomic characterization of SARS-CoV-2 in Brazil, promptly identifying novel variants to better respond and control its spread. 

In summary, our study corroborates the total virtual substitution of previous lineages by P.1 in Southern Brazil in COVID-19 cases sequenced in March 2020. Moreover, we confirmed various cases caused by the novel P.1.2 sublineage and placed its origin in the state of Rio Grande do Sul. The continuous evolution of the VOC P.1 is concerning, considering its clinical and epidemiological impact, and warrants enhanced genomic surveillance.

## 4. Materials and Methods

### 4.1. Sample Collection and Clinical Testing

Samples were obtained from Hospital da Brigada Militar patients, both admitted or visiting the emergency ward, from Porto Alegre, RS, Brazil. Nasopharyngeal swabs were collected and placed in saline solution. Samples were transported to the clinical laboratory (Laboratório Exame) and tested on the same day for SARS-CoV-2 using Real-Time Reverse-transcriptase Polymerase Chain Reaction (Charité RT-qPCR assays). The RTq-PCR assay used primers and probes recommended by the World Health Organization targeting the nucleocapsid (N1 and N2) genes [39]. Remnant samples were stored at −20 °C.

Between 9 March and 17 March 2021, all routinely tested samples of the clinical laboratory provenient of the Hospital da Brigada Militar patients and yielded positive RT-qPCR were selected. Subsequently, those positive clinical samples were submitted to a second RT-qPCR performed by BiomeHub (Florianópolis, Santa Catarina, Brazil), using the same protocol (charite-berlin). Only samples with quantification cycle (Cq) below 30 for at least one primer were submitted to the SARS-CoV-2 genome sequencing. In total, 56 patients who presented symptoms such as fever, cough, sore throat, dyspnea, anosmia, fatigue, diarrhea, and vomiting (moderate and severe clinical status) [40] were included in the study. 

### 4.2. RNA Extraction, Library Preparation, and Sequencing

Total RNAs were prepared as in the reference protocol [41] using SuperScript IV (Invitrogen, Carlsbad, CA, USA) for cDNA synthesis and Platinum Taq High Fidelity (Invitrogen, Carlsbad, CA, USA) for specific viral amplicons. Subsequently, cDNA was used for the library preparation with Nextera Flex (Illumina, San Diego, CA, USA) and quantified with Picogreen and Collibri Library Quantification Kit (Invitrogen, Carlsbad, CA, USA). The sequencing was performed on the Illumina MiSeq (Illumina, San Diego, CA, USA) 150 × 150 runs with 500xSARS-CoV-2 coverage (50–100 thousand reads/sample).

### 4.3. Quality Control and Consensus Calling

Quality control, reference mapping, and consensus calling were performed using an in-house pipeline developed by BiomeHub (Florianópolis, Santa Catarina, Brazil). Briefly, adapters were removed, and reads were trimmed by size = 150. Reads were mapped to the reference SARS-CoV-2 genome (GenBank accession number NC_045512.2) using Bowtie v2.4.2 (end-to-end and very-sensitive parameters) [42]. Mapping coverage and depth were retrieved using samtools v1.11 [43] (minimum base quality per base (Q) ≥ 30). Consensus sequences were generated using bcftools mpileup (Q ≥ 30; depth (d) ≤ 1000) combined with bcftools filter (DP > 50) and bcftools consensus v1.11 [44]. Coverage values for each genome were plotted using the karyoploteR v1.12.4 R package [45]. Finally, we assessed the consensus sequences quality using Nextclade v0.14.2 (https://clades.nextstrain.org/; accessed on 4 May 2021).

### 4.4. Mutation Analysis

SNPs and insertions/deletions in each sample were identified using snippy variant calling and core genome alignment pipeline v4.6.0 (https://github.com/tseemann/snippy; accessed on 4 May 2021), which uses FreeBayes v1.3.2 [46] to call variants and snpEff v5.0 [47] to annotate and predict their effects on genes and proteins. Genome map and SNP histogram were generated after running MAFFT v7.475 [48] alignment using the msastats.py script, and plotAlignment and plotSNPHist functions [49]. Sequence positions refer to GenBank RefSeq sequence (NC_045512.2), isolated and sequenced from an early case from Wuhan (China) in 2019.

We identified global virus lineages using the dynamic nomenclature implemented in Pangolin v2.3.8 [50] (https://github.com/cov-lineages/pangolin; accessed on 4 May 2021) and global clades and mutations using Nextclade v0.14.2 (https://clades.nextstrain.org/; accessed on 4 May 2021). We also used Pathogenwatch (https://pathogen.watch/; accessed on 4 May 2021) and Microreact [51] to explore mutations and lineages across time and geography initially.

### 4.5. Maximum Likelihood Phylogenomic Analysis

All available SARS-CoV-2 genomes (1,048,519 sequences) were obtained from GISAID on April 26, 2021 and combined with our 56 sequences to obtain a global representative dataset. These sequences were subjected to analysis inside the NextStrain ncov pipeline [52] (https://github.com/nextstrain/ncov; accessed on 4 May 2021). In this workflow, sequences were aligned using Nextalign v0.1.6 (https://github.com/neherlab/nextalign; accessed on 4 May 2021). In the initial filtering step, short and low-quality sequences or those with incomplete sampling dates were excluded. Uninformative sites and ends (100 positions in the beginning and 50 in the end) were also masked from the alignment. Genetically closely related genomes to our focal subset were selected, prioritizing sequences geographically closer to Brazil’s RS state. The maximum likelihood (ML) phylogenetic tree was built using IQ-TREE v2.1.2 [53], employing the general time-reversible (GTR) model with unequal rates and base frequencies [54]. The tree’s root was placed between lineage A and B (Wuhan/WH01/2019 and Wuhan/Hu-1/2019 representatives), and sequences that deviate more than four interquartile ranges from the root-to-tip regression of genetic distances against sampling dates were removed from the analysis. A time-scaled ML tree was generated with TreeTime v0.8.1 [55] under a strict clock and a skyline coalescent prior with a mean rate of 8 × 10^−4^ substitutions per site per year. Finally, clades and mutations were assigned and geographic movements inferred. The results were exported to JSON format to enable interactive visualization through Auspice. 

Additionally, as P.1 sequences mostly represent our dataset, we downloaded all complete and high-quality global genomes assigned to P.1 PANGO lineage (4499 sequences) submitted until 26 April 2021. These sequences were aligned using MAFFT v7.475, the ends of the alignment (300 in the beginning and 500 in the end) were masked, and the ML tree was built with IQ-TREE v2.0.3 using the GTR + F + R3 nucleotide substitution model as selected by the ModelFinder [56]. Branch support was calculated using the Shimodaira–Hasegawa approximate likelihood ratio test (SH-aLRT) [57] with 1000 replicates.

Local sequences were classified according to the following scheme: monophyletic clades composed by one local genome were classified as “isolated”, while clades composed by 2 < genomes < 4 were considered “clusters”, and, if ≥ 4 local genomes were represented, we assigned a “clade” designation.

ML trees were inspected in TempEst v1.5.3 [58] to investigate the temporal signal through regression of root-to-tip genetic divergence against sampling dates. For the P.1 ML tree, samples with missing days of the collection were filled with the 15th day of the month. ML and time-resolved trees were visualized using FigTree v1.4.4 (http://tree.bio.ed.ac.uk/software/figtree/; accessed on 4 May 2021) and ggtree R package v2.0.4 [59].

### 4.6. Discrete Bayesian Phylogeographic and Phylodynamic Analysis

Considering the four identified clades composed of ≥4 sequences from this study, we extracted the clade members using the caper R package v1.0.1 [60]. Clade-specific ML trees and root-to-tip regression assignments were generated as described above. Evolutionary parameter estimates and spatial diffusion were assessed separately for each clade using a Bayesian Markov Chain Monte-Carlo (MCMC) approach implemented in BEAST v10.4 [61]. The BEAGLE library [62] was used to enhance computational time. Time-stamped Bayesian trees were generated using the HKY + Γ nucleotide model [63], a strict molecular clock model with a Continuous Time Markov Chain (CTMC) rate reference prior [64] (mean rate = 8 × 10^−4^) and a non-parametric skygrid tree prior [65] with grid points defined by the approximate number of weeks spanned by the duration of the phylogeny.

The MCMC chains were run in duplicates for at least 50 million generations, and convergence was checked using Tracer v1.7.1 [66]. Log and tree files were combined using LogCombiner v1.10.4 to ensure stationarity and good mixing after removing 10% as burn-in. Maximum clade credibility (MCC) was generated using TreeAnnotator v1.10.4 [61]. Viral migrations were reconstructed using a reversible discrete asymmetric phylogeographic model [67] to infer the locations of the internal nodes of the tree. A discretization scheme with a resolution of different Brazilian states and other countries was applied. Location diffusion rates were estimated using the Bayesian stochastic search variable selection (BSSVS) [67] procedure employing Bayes factors to identify well-supported rates.

### 4.7. Geoplotting

Geographical maps and general plots were generated using R v3.6.1 [68], and the ggplot2 v3.3.2 [69], geobr v.1.4 [70], and sf v0.9.8 [71] packages. For the discrete phylogeographic analysis, SpreaD3 v0.9.7.1 software [72] was used to map spatiotemporal information embedded in MCC trees.

## Figures and Tables

**Figure 1 pathogens-10-00988-f001:**
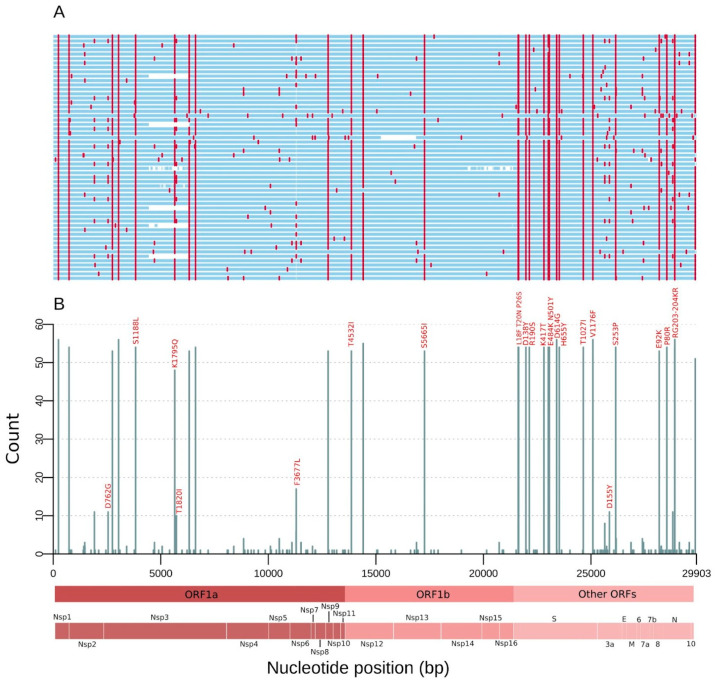
Mutations of the SARS-CoV-2 genomes from RS state, Southern Brazil sampled in March 2021. (**A**) Genome map for the 56 genomes sequenced. Nucleotide substitutions are colored in red and blank regions represent low sequencing coverage. (**B**) Count of single nucleotide polymorphisms (SNPs) per SARS-CoV-2 genome position along the 56 genomes. These mutations are corresponding to the red lines in (**A**), and only missense substitutions represented by >10 sequences have their respective amino acid changes indicated above the bars. Main open reading frames (ORFs) and SARS-CoV-2 proteins are indicated at the bottom to allow a rapid visualization of the viral proteins affected.

**Figure 2 pathogens-10-00988-f002:**
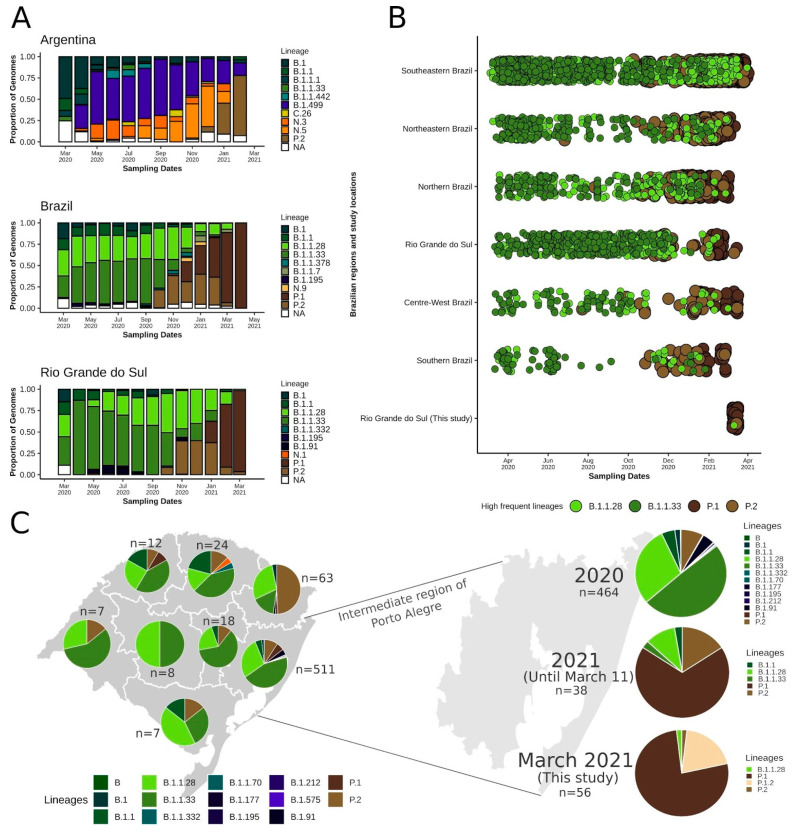
Distribution of SARS-CoV-2 lineages across time in Argentina, other Brazilian regions, and RS state. The other bordering country (Uruguay) was not included due to the limited number of samples available (n = 135). (**A**) Per month lineage distribution in Argentina, all of Brazil, and RS state. Only the 10 most prevalent lineages were considered. (**B**) Timeline showing the distribution of the most prevalent lineages until the end of 2020 (B.1.1.28 and B.1.1.33) and from the end of 2020 onward (P.1 and P.2) for the five Brazilian regions, RS state, and considering only this study (March 2021). (**C**) Map of RS state divided into eight intermediate regions, as defined by IBGE, displaying the proportion of the lineages in each area. The region of Porto Alegre was amplified and the lineage frequencies from 2020, 2021 and the present study are presented on the right. NA, other lineages.

**Figure 3 pathogens-10-00988-f003:**
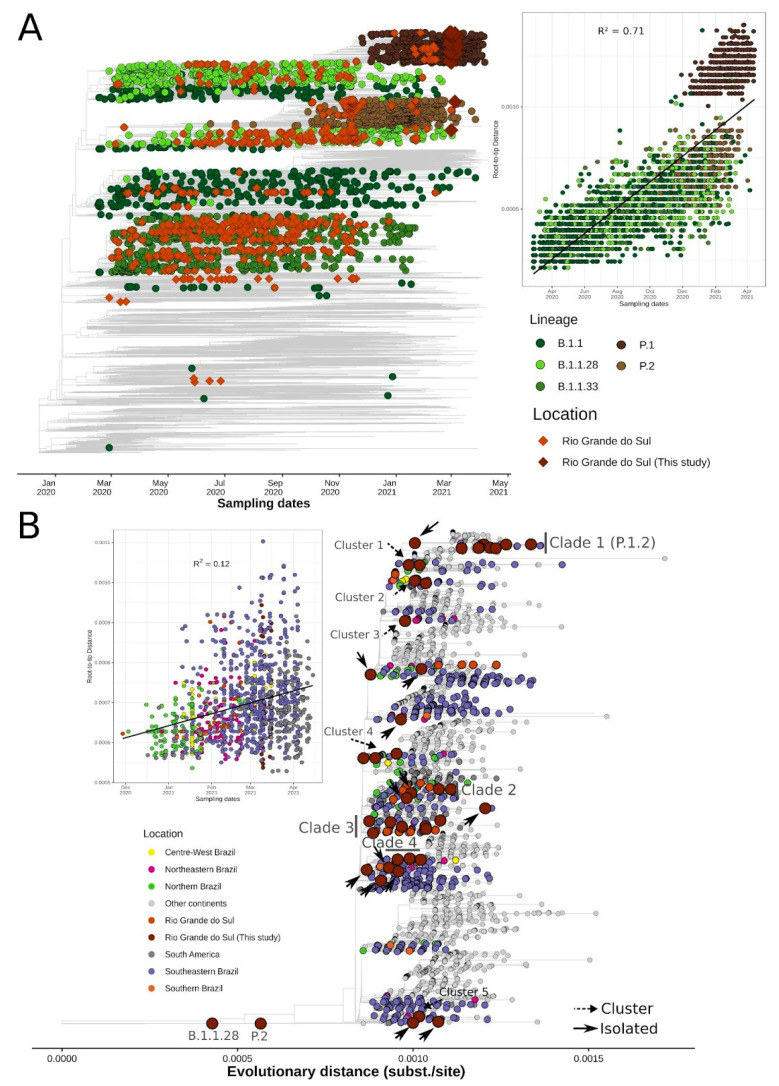
Phylogenetic analysis of genomes sequences in RS state in a global context. (**A**) Time-resolved ML tree of 8635 global representative SARS-CoV-2 genomes. Circles represent global sequences belonging to the five most abundant lineages in RS state that passed quality control criteria: B.1.1 (n = 32), P.1 (n = 83), P.2 (n = 83), B.1.1.28 (n = 203), and B.1.1.33 (n = 286). Diamonds represent RS genomes (available in GISAID and sequenced in this study). Root-to-tip regression is represented on the right of the tree. (**B**) ML tree of 4499 SARS-CoV-2 genomes belonging to the P.1 lineage. Tips are colored by Brazilian regions, South America, or other continents. Introductions, clusters, and clades are annotated in the tree (see Methods). Root-to-tip regression is depicted on the left of the tree and sequences from “other continents” were dropped to improve visualization.

**Figure 4 pathogens-10-00988-f004:**
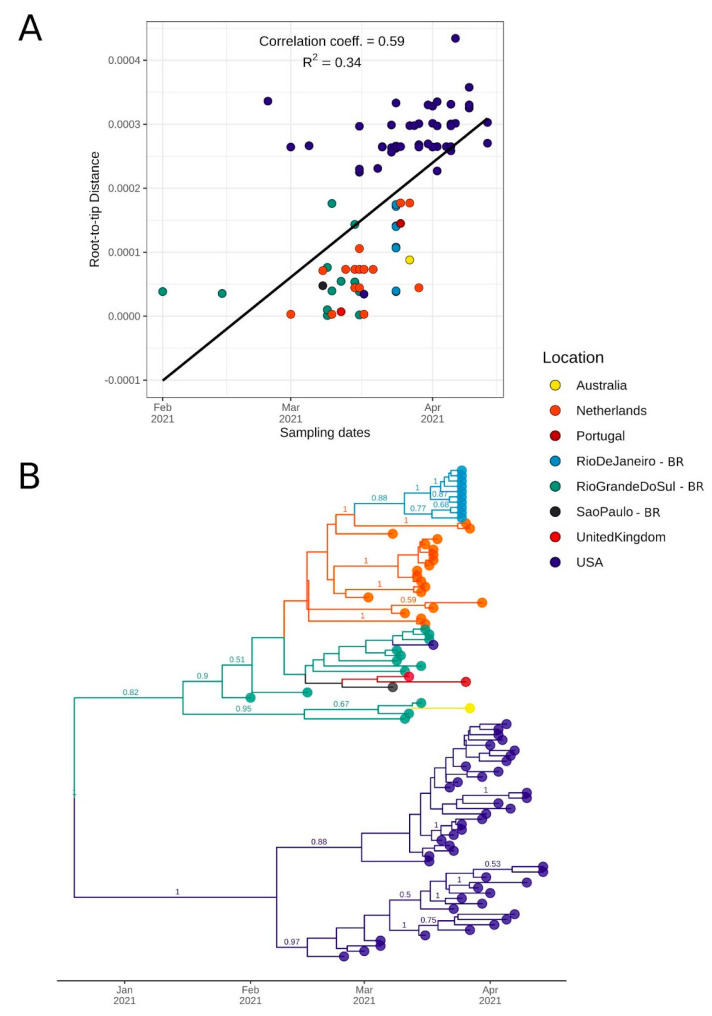
Bayesian discrete asymmetric phylogeographic analysis of the identified Clade 1 (lineage P.1.2). (**A**) Root-to-tip regression of genetic distances and sampling dates for Clade 1. Correlation coefficient and R squared are depicted above the graph. (**B**) MCC tree of the 93 sequences included in this analysis up to 26 April 2021 (82 from GISAID and 11 from this study). Numbers above branches represent the posterior probability of each branch. Only posteriors > 0.5 are shown. Circles indicate countries outside Brazil and Brazilian states (BR suffix).

**Figure 5 pathogens-10-00988-f005:**
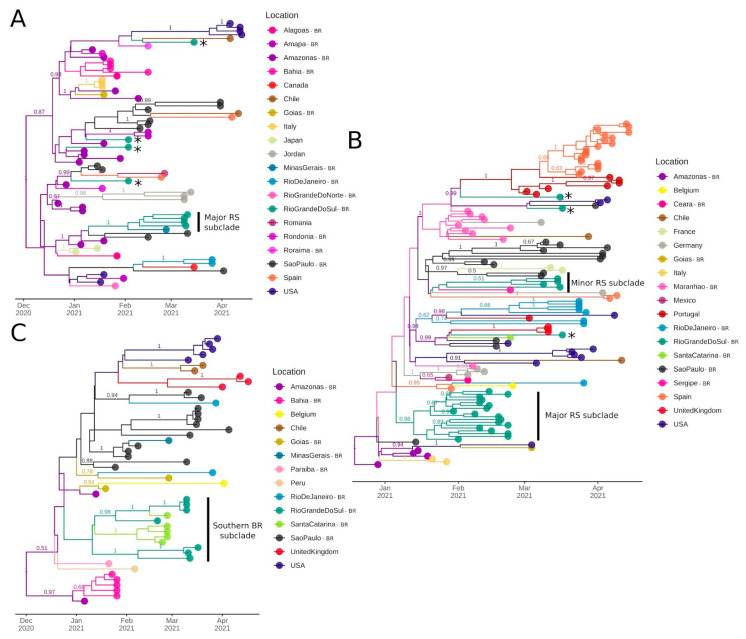
Bayesian discrete asymmetric phylogeographic analysis of the identified Clades 2–4 up to 26 April, 2021. (**A**) Clade 2: MCC tree of the 71 sequences included in this analysis. (**B**) Clade 3: MCC tree of the 122 genomes included in this analysis. (**C**) Clade 4: MCC tree of the 50 sequences included in this analysis. For all MCC trees, numbers above branches represent the posterior probability of each branch. Only posteriors > 0.5 are shown. Asterisks represent potential introductions in RS state and subclades cited in the text are indicated. Circles indicate countries outside Brazil and Brazilian states (BR suffix).

**Table 1 pathogens-10-00988-t001:** Epidemiological characteristics of the 56 sequenced samples from Rio Grande do Sul, Southern Brazil.

Study ID	GISAID ID	Cycle Threshold	Municipality of Residence	Gender	Age Group	Lineage	Contact with Confirmed Case
(HBM-RS)	(EPI_ISL_)						
39468	2139494	16	São Leopoldo	Male	30–39	P.1	Yes
39469	2139495	19	Porto Alegre	Female	20–29	P.1.2	Yes
39470	2139496	19	Porto Alegre	Male	60–69	P.1	No
39471	2139497	18	Porto Alegre	Male	20–29	P.1	Yes
39472	2139498	17	Gravataí	Male	30–39	P.1	No
39473	2139499	26	Cachoeira do Sul	Female	20–29	P.1	Yes
39474	2139500	18	Gravataí	Male	30–39	P.1	Yes
39475	2139501	18	Porto Alegre	Female	20–29	P.1	No
39476	2139502	15	Porto Alegre	Male	20–29	P.1	Yes
39477	2139503	21	Porto Alegre	Male	30–39	P.1	Yes
39478	2139504	15	Cachoeira do Sul	Male	30–39	P.1	Yes
39479	2139505	22	Porto Alegre	Male	50–59	P.1	Yes
39480	2139506	17	Novo Hamburgo	Male	40–49	P.1	Yes
39481	2139507	14	Porto Alegre	Female	70–79	P.1	Yes
39482	2139508	14	Porto Alegre	Female	80–89	P.1.2	No
39483	2139509	13	Gravataí	Male	30–39	P.1	Yes
39484	2139510	20	Porto Alegre	Male	20–29	P.1	Yes
39485	2139511	16	Porto Alegre	Male	50–59	P.1	Yes
39486	2139512	27	Porto Alegre	Male	30–39	P.2	No
39487	2139513	14	São Sebastião do Caí	Male	40–49	P.1.2	Yes
39488	2139514	28	Santo Antônio da Patrulha	Male	70–79	P.1	Yes
39489	2139515	27	Porto Alegre	Female	20–29	P.1.2	Yes
39490	2139516	18	Porto Alegre	Male	20–29	P.1	Yes
39491	2139517	15	Alvorada	Female	20–29	B.1.1.28	Yes
39492	2139518	17	Gravataí	Female	30–39	P.1	Yes
39493	2139519	22	Canoas	Male	30–39	P.1.2	Yes
39494	2139520	17	Porto Alegre	Female	30–39	P.1	No
39495	2139521	17	Porto Alegre	Male	30–39	P.1	Yes
39496	2139522	17	Canoas	Female	30–39	P.1	Yes
39497	2139523	21	Porto Alegre	Male	40–49	P.1.2	Yes
39498	2139524	20	Porto Alegre	Female	40–49	P.1	Yes
39499	2139525	22	Portão	Male	30–39	P.1	Yes
39500	2139526	11	Porto Alegre	Male	20–29	P.1.2	Yes
39501	2139527	14	Santa Maria	Male	20–29	P.1.2	Yes
39502	2139528	21	Porto Alegre	Male	30–39	P.1	Yes
39503	2139529	16	Porto Alegre	Male	30–39	P.1	No
39504	2139530	21	Gravataí	Male	40–49	P.1	Yes
39505	2139531	13	Porto Alegre	Male	30–39	P.1.2	Yes
39506	2139532	23	Porto Alegre	Female	40–49	P.1	Yes
39507	2139533	28	Canoas	Female	30–39	P.1	Yes
39508	2139534	22	Porto Alegre	Male	20–29	P.1	Yes
39509	2139535	23	Alvorada	Male	20–29	P.1	Yes
39510	2139536	19	Canoas	Male	50–59	P.1.2	Yes
39511	2139537	22	Porto Alegre	Male	30–39	P.1	No
39512	2139538	25	Cachoeira do Sul	Female	40–49	P.1	Yes
39513	2139539	23	Santa Maria	Male	40–49	P.1	Yes
39514	2139540	15	Porto Alegre	Male	30–39	P.1	No
39515	2139541	21	Porto Alegre	Male	20–29	P.1	Yes
39516	2139542	28	Porto Alegre	Male	50–59	P.1	Yes
39517	2139543	17	Sapiranga	Male	30–39	P.1	Yes
39518	2139544	17	Porto Alegre	Male	30–39	P.1.2	Yes
39519	2139545	23	Porto Alegre	Male	20–29	P.1	Yes
39520	2139546	15	Campo Bom	Male	20–29	P.1	Yes
39521	2139547	15	Porto Alegre	Male	20–29	P.1	Yes
39522	2139548	21	Porto Alegre	Male	50–59	P.1	Yes
39523	2139549	18	Porto Alegre	Male	20–29	P.1	Yes

All samples were nasopharyngeal swabs collected during 9–17 March 2021, from residents of RS state. Study ID: Study identifier only known by study investigators. The Ct values are related to the first RT-qPCR conducted at Laboratório Exame.

**Table 2 pathogens-10-00988-t002:** Detailed description and frequency of mutations found in our 56 sequences compared with all Brazilian P.1 sequences until 26 April 2021.

Genomic Position	Effect	Amino Acid Change	Gene/Region	Product	Frequency Our Study (%)	Frequency in Brazilian’s P.1 (%)
C241T	Intergenic	NA	5′ UTR	NA	100.0	97.2
T733C	Synonymous	D156D	ORF1ab	Leader Protein	96.4	99.9
C1912T	Synonymous	S549S		nsp2	19.6	1.4
A2550G	Missense	D762G			19.6	1.5
C2749T	Synonymous	D828D		nsp3	94.6	99.7
C3037T	Synonymous	F924F			100.0	99.9
**C3828T**	**Missense**	**S1188L**			96.4	95.3
**A5648C**	**Missense**	**K1795Q**			85.7	100.0
C5724T	Missense	T1820I			17.9	2.3
A6319G	Synonymous	P2018P			87.5	99.5
A6613G	Synonymous	V2116V			96.4	99.8
T11296G	Missense	F3677L		nsp6	30.4	8.2
C12778T	Synonymous	Y4171Y		nsp9	94.6	98.9
C13860T	Missense	T4532I		RdRp	94.6	99.8
C14408T	Synonymous	L4715L			98.2	96.8
G17259T	Missense	S5665I		Helicase	94.6	99.7
**C21614T**	**Missense**	**L18F**	S	Surface Glycoprotein	96.4	99.9
**C21621A**	**Missense**	**T20N**			94.6	99.8
**C21638T**	**Missense**	**P26S**			96.4	99.1
**G21974T**	**Missense**	**D138Y**			96.4	100.0
**G22132T**	**Missense**	**R190S**			96.4	98.4
**A22812C**	**Missense**	**K417T**			96.4	83.4
**G23012A**	**Missense**	**E484K**			96.4	99.9
**A23063T**	**Missense**	**N501Y**			96.4	99.8
A23403G	Missense	D614G			100.0	97.7
**C23525T**	**Missense**	**H655Y**			96.4	100.0
**C24642T**	**Missense**	**T1027I**			96.4	99.9
G25088T	Missense	V1176F			100.0	99.9
G25855T	Missense	D155Y	ORF3a	ORF3a Protein	19.6	1.6
**T26149C**	**Missense**	**S253P**			94.6	98.7
**G28167A**	**Missense**	**E92K**	ORF8	ORF8 Protein	94.6	99.8
**C28512G**	**Missense**	**P80R**	N	Nucleocapsid Phosphoprotein	96.4	98.3
C28789T	Synonymous	Y172Y			19.6	1.3
AGTAGGG 28877–28883 TCTAAAC	Missense	RG203-204KR			96.4	99.8

Original bases or amino acids are represented before the genomic coordinate, while the mutated ones are presented after. Only mutations observed in more than 10 genomes from this study are shown. P.1 lineage-defining mutations are highlighted in bold. P.1.2 (new lineage) defining replacements are underlined and marked with gray background color. UTR, untranslated region; ORF, open reading frame; S, spike; N, nucleocapsid; nsp, nonstructural protein; RdRp, RNA-dependent RNA polymerase.

## Data Availability

Full tables acknowledging the authors and corresponding labs submitting sequencing data used in this study can be found in Files S3 and S4. Consensus genomes generated in this study were deposited in the GISAID database under Accession IDs: EPI_ISL_2139494 to EPI_ISL_2139549. Data and code used to reproduce the results presented are available in Appendix A.

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
