# Peer review of "Predominance of the SARS-CoV-2 Lineage P.1 and Its Sublineage P.1.2 in Patients from the Metropolitan Region of Porto Alegre, Southern Brazil in March 2021"

_pathogens, 2021, doi:10.3390/pathogens10080988_

Round 1

Reviewer 1 Report

In this manuscript, Bonetti franceschi and co-authors report a very detailed analysis of SARS-CoV-2 genome sequences obtained from samples collected in the region of Rio Grande Do Sul, Southern Brasil. They assess the prevalence of VOC P.1 and the new P.1.2 and finally put these sequences in a wider context of the P.1 phylogeny, concluding that there is substantial diversity to justify multiple introductions of the variant in the region and dating the events of divergence using a Bayesan approach.

The results are interesting and well presented, the methods are clearly explained. 

I have only one major concern which is related to the length of the manuscript, which is too much in my opinion. In particular the discussion section must be reduced, it is not a review of the literature and only the information that is significant for the present work must be reported and discussed.

For example, from line 399 to line 467: this is a detailed report of the effect of most of the mutations present in the P.1 VOC, that is not necessary.

I suggest to the authors to significantly reduce the paper length and to summarize the relevant information, thus helping the reader to focus on the most important results.

Minor points:

  • line 107: were --> was
  • line 135: I cannot reproduce the 175 count, I get 172, what am I missing?
  • lines 147-150: the deletion is rarely found because of coverage filtering. Why this is not affecting also the detection of other mutations?
  • Figure 1: there are mutations reported in A that seem to reach almost 100% of frequency, but in B there are no mutations that reach 60%
  • line 175: two (in letters). Sometimes numbers are in letters, sometimes in digits, please make them uniform
  • lines 177-179: please rephrase
  • lines 204-206: please rephrase
  • line 216: missing closing parenthesis
  • lines 390-392: please rephrase
  • line 417: vaccine development --> vaccine
  • line 450: dissemination is repeated multiple times
  • line 560: exploit --> explore?
  • line 571: "promptly" should be moved before "identifying" 
  • lines 643-644: under...under
  • 669-671: estimates...estimated; using...using

Reviewer 2 Report

This paper is great. I can usually find a few things wrong/fixes to input but honestly this was a great read. Awesome figures, really well researched. Your discussion is a little long and felt more like a review paper

Reviewer 3 Report

Franceschi et al. we sequenced 56 viral samples from the metropolitan region of Porto Alegre to perform a genomic characterization. They found that majority of the samples belonged to the P.1 38 lineage and approximately 20% of them were assigned as the novel a P.1-derived sublineage P.1.2. The authors further concluded that the sequences from this study were allocated in distinct branches of the P.1 phylogeny, suggesting multiple introductions in RS still in 2020 and placing this state as a potential diffusion and emergence core of P.1-derived clades.

Major concerns:

  1. The 56 samples examined in this study was collected from a very short period time of time (March 9-17th ) 2021. How did the author find evidence that “multiple introductions in RS still in 2020”? In line 586, please add the year of sample collection. As mentioned by the authors in line 561, since the sample size is low ( and collected within a short period of time) and not necessarily representative of the RS state, this decreased the significance and reliability of this study.  some of the conclusions and statements in the manuscript need to be revised to reflect this weakness.   
  2. Line 98-99, “In the RS state, the most common lineages identified to May 2021 still are B.1.1.33 (n=290) and B.1.1.28 (n=238). “, please add reference to this statement. Please compare this dataset, which include more than 500 samples, with the current one. how this statement was different from the current study?
  3. Part of the novelty in this is the sublineage P1.2. Please make a new table to list the major difference between P1 vs P1.2.  

Minor concerns:

  1. Please add the WHO labels of the SARS-Cov-2 variants, eg, alpha, beta etc.
  2. Reference #16 is not English.

Round 2

Reviewer 3 Report

The concerns have been addressed.